# UMZero: A Unified CNN-Mamba Framework for Zero-Shot learning

## Abstract

In zero-shot learning (ZSL), accurately recognizing unseen classes relies heavily on deeply understanding both local details and global correlations between visual and semantic modalities. While prior methods have leveraged CNNs for local feature extraction or utilized Transformers and Mamba models to capture global contextual information, these approaches often lack an integrated mechanism to jointly model both aspects. This limitation hampers their ability to fully exploit complex cross-modal interactions. To overcome this challenge, we propose UMZero, a novel hybrid ZSL framework that synergistically combines a pretrained CNN with a state space model (SSM), effectively uniting fine-grained local feature extraction with long-range dependency modeling. UMZero is composed of three core modules: the High-order Global Aggregator (HGA), the Mamba Interaction Module (MIM), and a prototype learning unit. Specifically, the HGA enhances feature expressiveness by capturing high-order statistical dependencies across channels; the MIM enables deep cross-modal fusion and alignment by jointly modeling local-global interactions and supporting bidirectional information flow between modalities; and the prototype learning module constructs a semantically structured embedding space, promoting compact intra-class and discriminative inter-class representations. Extensive evaluations on several well-established ZSL datasets confirm that UMZero surpasses current state-of-the-art methods, delivering superior performance and demonstrating robust generalization capabilities. The source code will be publicly released upon paper acceptance.

## 1 Introduction

Zero-shot learning (ZSL) (Pourpanah et al., 2022) addresses the problem of recognizing categories unseen during training by leveraging auxiliary semantic information—such as attributes, word embeddings, or textual descriptions (Lampert et al., 2013)—to bridge seen and unseen classes. Mainstream methods (Huynh & Elhamifar, 2020; Xu et al., 2022; Chen et al., 2023a; Naeem et al., 2023; Chen et al., 2024c; Hou et al., 2025) primarily construct vision–semantic alignment mechanisms so that knowledge learned on seen classes can transfer to novel ones. Despite these advances, learning highly discriminative visual and semantic representations remains a central challenge in ZSL. With the advent of deep architectures, ZSL performance has improved markedly. CNNs (He et al., 2016) excel at extracting fine-grained local features but have limited capacity for capturing long-range dependencies due to their finite receptive fields (Luo et al., 2016). ViTs (Dosovitskiy et al., 2020), by contrast, model global context via self-attention but incur substantial computational cost (Habib et al., 2023). Recently, state space models (SSMs) (Zhu et al., 2024)—and in particular the Mamba architecture (Dao & Gu, 2024)—have demonstrated efficient long-range context modeling with low overhead. Its visual variant, Vision Mamba (Zhu et al., 2024), has achieved strong results on image classification and point-cloud tasks by effectively capturing global dependencies. Figure 5 illustrates these trade-offs: CNNs (a) capture local textures, ViTs (b) and SSMs (c) model global context, and SSMs do so with far less computational expense than ViTs. Inspired by these findings, ZeroMamba (Hou et al., 2025) is the first to replace CNN or ViT backbones with pretrained Vision Mamba modules for end-to-end ZSL, yielding encouraging initial results. However, ZeroMamba has two key limitations: (1) Vision Mamba, despite its global modeling prowess, lacks fine-grained local discrimination in the absence of CNN features; and (2) existing ZSL models primarily rely

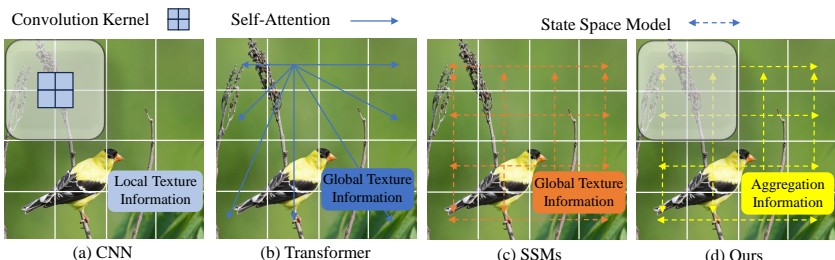

Figure 1: Motivation illustration. (a) CNN, (b) ViT, (c) State Space Model (SSM), and (d) Our proposed UMZero.

on semantic information to guide the update of visual representations, lacking a bidirectional interaction mechanism between vision and semantics. As a result, semantic prototypes cannot be dynamically refined based on visual feedback, making it difficult to bridge the representational gap between semantic and visual spaces.

To overcome the above limitations, we propose UMZero, which optimizes the fusion strategy by embedding the local information extracted by CNNs into the global contextual representations captured by SSMs, thereby enhancing the expressiveness of local details within the overall representation (as illustrated in Fig. 5(d)). The overall architecture is shown in Fig. 6 First, the High-order Global Aggregator (HGA) significantly improves feature representation by modeling high-order statistical relationships among channels. Furthermore, we introduce the Mamba Interaction Module (MIM), which combines the fine-grained local feature extraction capability of CNNs with the powerful global semantic modeling ability of Mamba to generate high-quality visual representations that capture both local details and global structure. Meanwhile, dynamic attention-based interactions between visual and semantic information facilitate semantic-aware visual region modeling and visually enhanced semantic prototype refinement. Finally, our prototype learning module introduces a prototype contrastive loss to enhance inter-class separability and intra-class compactness in the embedding space, thereby improving the model's discriminative ability on unseen classes. The main contributions of this work are summarized as follows:

- We propose UMZero, the first hybrid ZSL framework that integrates CNN and Mamba architectures, effectively combining local attribute extraction with global semantic modeling to enhance generalization ability.
- We introduce the High-order Global Aggregator (HGA) to capture high-order statistical relationships among channels, significantly enhancing feature representation quality.
- We propose the Mamba Interaction Module (MIM) to enable dynamic bidirectional cross-modal interactions for precise visual-semantic alignment.
- We design a prototype contrastive loss to improve intra-class compactness and inter-class separability, optimizing the structure of the embedding space.

## 2 RELATED WORK

Zero-shot learning (ZSL) aims to recognize novel categories through visual-semantic interaction, thereby reducing the reliance on annotated samples inherent in conventional supervised approaches. Existing ZSL approaches can be broadly classified into generative and embedding-based methods (Chao et al., 2016), which typically employ CNNs or ViTs to extract visual features and utilize class-level attributes as the semantic source. A central challenge in ZSL is to enhance visual–semantic consistency and construct robust cross-modal representations. Early CNN-based methods (Huynh & Elhamifar, 2020; Chen et al., 2021) often map global visual features into the semantic space, but fail to capture fine-grained discriminative cues. To address this, recent studies (Liu et al., 2024; Chen et al., 2022b; Liu et al., 2021) incorporate attention mechanisms to mine salient regions: AREN (Xie et al., 2019) and SGMA (Zhu et al., 2019) apply occlusion strategies to emphasize regional information; DAZLE (Huynh & Elhamifar, 2020) and MSDN (Chen et al., 2022c) use dense attention to precisely localize attribute-specific patches; APN (Xu et al., 2022) and

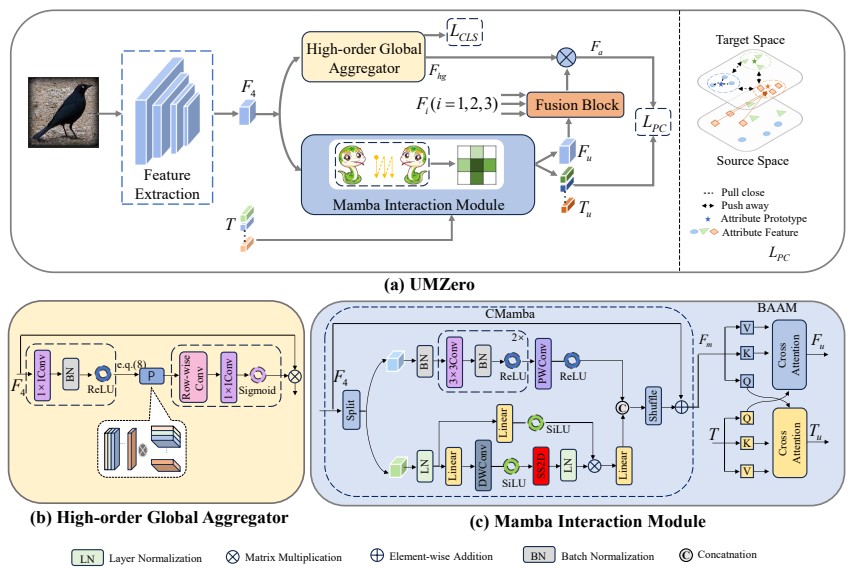

**(a) UMZero**

**(b) High-order Global Aggregator**  **(c) Mamba Interaction Module**

LN Layer Normalization  ⊗ Matrix Multiplication  ⊕ Element-wise Addition  BN Batch Normalization  Ⓒ Concatnation

Figure 2: The framework of UMZero. It uses a pre-trained ResNet-101 to extract multi-level features, with the HGA refining global representations. The MIM enhances local feature aggregation and adaptively optimizes visual and semantic features by fusing shallow information to improve semantic completeness. Finally, a prototype contrastive loss guides the construction of highly discriminative attribute-level representations.

DPPN (Wang et al., 2021) enhance attribute modeling via prototype construction and refinement; and TransZero (Chen et al., 2022b) leverages a transformer-based query mechanism to guide semantic alignment. While these methods strengthen visual-semantic alignment, they often overlook global semantic structures. The introduction of ViT architectures, renowned for modeling long-range dependencies (Radford et al., 2021; Chen et al., 2023c; Liu et al., 2025; Yue et al., 2025), has further advanced ZSL. ViT backbones have gradually supplanted CNNs to enrich feature expressiveness. For example, PSVMA (Liu et al., 2023) and ZSLViT (Chen et al., 2024b) boost visual discriminability through ViT, while I2MVFormer (Naeem et al., 2023) and DUET (Chen et al., 2023c) integrate large language models (LLMs) to enrich semantic understanding and improve cross-modal alignment. However, the quadratic complexity of transformer self-attention limits their deployment in resource-constrained scenarios. Recently, Mamba, a state–space model (SSM)–based architecture, has emerged as a promising alternative. By employing a dynamic weighting mechanism with linear computational complexity, Mamba maintains strong global modeling capacity while significantly reducing inference overhead. ZeroMamba (Hou et al., 2025) uses a pre-trained Vision Mamba backbone for ZSL and demonstrates encouraging initial performance. Nevertheless, its implicit treatment of local regions constrains its fine-grained attribute modeling. However, its performance in fine-grained attribute modeling remains limited. In contrast, CNNs excel at capturing local structures and details, making them a complementary component to Mamba's global modeling.

Motivated by these observations, we propose a hybrid ZSL framework that integrates CNNs and Mamba. Our method combines the local perceptual strengths of CNNs with the global contextual modeling of Mamba to enhance discriminative attribute learning. By fully leveraging both local details and global context, the proposed method delivers a more efficient and effective solution for ZSL.

## 3 METHODOLOGY

In UMZero (Fig. 6), we integrate a hybrid CNN–Mamba architecture for ZSL. This section first formalizes the ZSL task, then provides a brief overview of state-space models. We subsequently detail the core design and implementation of our method, and conclude with the model optimization process and inference strategy under the ZSL setting.

### 3.1 Notation and Problem Settings.

In ZSL, the seen dataset is defined as $\mathcal{D}_s = \{(x_i, y_i) \mid x_i \in \mathcal{X}_s, y_i \in \mathcal{Y}_s\}$ with $\mathcal{C}_s$ classes, while the unseen dataset $\mathcal{D}_u = \{(x_i, y_i) \mid x_i \in \mathcal{X}_u, y_i \in \mathcal{Y}_u\}$ contains $\mathcal{C}_u$ disjoint classes, i.e., $\mathcal{C}_s \cap \mathcal{C}_u = \emptyset$, where $x_i$ represents an image and $y_i$ its corresponding label. Conventional ZSL (CZSL) trains on $\mathcal{D}_{tr} \subseteq \mathcal{D}_s$ to classify unseen classes, while generalized ZSL (GZSL) aims to recognize both seen and unseen classes from $\mathcal{C} = \mathcal{C}_s \cup \mathcal{C}_u$. To align visual and semantic information, class and attribute labels are mapped into a visual embedding space via an encoder (three fully connected layers with ReLU), generating class prototype $P^c \in \mathbb{R}^{M \times C}$ and attribute prototype $T \in \mathbb{R}^{K \times C}$, where $M$ is the number of classes, $C$ is the embedding dimension, and $K$ is the number of attributes.

### 3.2 Preliminaries

The preliminaries of Structured State Space Models (SSMs) and the corresponding model illustration are provided in the Appendix A.1.

### 3.3 UMZero

As illustrated in Fig.6, the UMZero framework consists of three main components: the HGA, the MIM module, and the prototype learning module. It utilizes a pre-trained ResNet-101 to extract multi-level visual features. To enhance feature quality, a dual-branch design is applied: HGA performs high-order modeling on visual features to enhance global representations, while MIM strengthens local feature aggregation and contextual perception, and adaptively filters and refines both visual and semantic features. The refined features are then fused with shallow features to improve semantic completeness. Finally, the fused features and the output of HGA are used together to construct attribute-level representations, which are guided by a prototype contrastive loss to enhance attribute discriminability and the model's generalization ability.

#### 3.3.1 High-order Global Aggregator(HGA)

Given a visual feature map $F_4 \in \mathbb{R}^{C \times H \times W}$ (i.e., the output of the final layer) generated by the feature extractor, where $C$ denotes the number of channels and $H$ and $W$ represent the spatial dimensions, directly using the raw features often introduces redundancy and overlooks high-order dependencies among channels. To address this, we propose the High-order Global Aggregator (HGA)(Fig.6(b)), which models second-order statistical relationships across channels to enhance feature compactness and discriminative power.

First, a $1 \times 1$ convolution is applied to reduce the channel dimension from $C$ to $C'$, improving efficiency while preserving key information:

$$f = \text{ReLU}\big(\text{BN}(\text{Conv}_{1\times1}(F_4))\big) \in \mathbb{R}^{C' \times H \times W}, \tag{1}$$

Next, mean-centering is performed on each channel:

$$\tilde{f} = f - \bar{f}, \quad \bar{f} = \frac{1}{HW} \sum_{i=1}^{H} \sum_{j=1}^{W} f_{[:,i,j]}, \tag{2}$$

The feature map is then flattened into $\tilde{f} \in \mathbb{R}^{C' \times N}$ where ($N = H \times W$), and a covariance matrix is computed to capture second-order channel statistics:

$$P = \frac{1}{N} \tilde{f} \cdot \tilde{f}^{\top} \in \mathbb{R}^{C' \times C'}, \tag{3}$$

Row-wise grouped convolution followed by a $1 \times 1$ convolution and Sigmoid activation is applied to $P$ to generate a channel attention vector $w \in \mathbb{R}^C$:

$$w = \sigma\big(\text{Conv}_{1\times1}(\text{RowConv}(P))\big), \tag{4}$$

The attention vector is used to reweight the original feature map $F_4$ channel-wise:

$$F_{hg} = F_4 \otimes w, \tag{5}$$

### 3.3.2 MAMBA INTERACTION MODULE(MIM)

The MIM consists of the Convolution-Mamba(CMamba) and Bidirectional Aware Attention Module(BAAM) components, designed to capture multi-directional spatial dependencies and fine-grained local features in images while simultaneously updating visual and semantic representations, as shown in Fig.6(c).

**Convolution-Mamba Module(CMamba)** It first splits the input feature map $F_4$ equally along the channel dimension:

$$F_4 = [F^{(1)}, F^{(2)}], \quad F^{(1)}, F^{(2)} \in R^{H \times W \times \frac{C}{2}}, \tag{6}$$

Here, $F^{(1)}$ is passed into the convolutional(the upper) branch, while $F^{(2)}$ is fed into the Mamba(the lower) branch, enabling joint extraction of local details and global context.

Convolutional Branch focuses on modeling local structural features. The input is first normalized by Batch Normalization to stabilize feature distribution. It then passes through two identical submodules, each consisting of a $3 \times 3$ convolution, normalization, and activation function, progressively extracting fine-grained local features. Finally, a pointwise $(1 \times 1)$ convolution followed by an activation function is applied, producing the output of the convolutional branch as:

$$F_c{}^{(1)} = R \left( L \left( (C_3 N_o R)^2 \left( N_o(F^{(1)}) \right) \right) \right), \tag{7}$$

where $N_o$ denotes layer normalization, $C_3$ denotes a $3 \times 3$ convolution, $R$ represents an ReLU activation function, and $L$ denotes a linear transformation.

Mamba Branch is designed to model global contextual dependencies and consists of two parallel sub-paths: the Linear-SiLU path and SS2D path. The input $F^{(2)}$ is first processed with Layer Normalization to produce $F' = N_o(F^{(2)})$ , which is then passed through the two paths:

- **SS2D Path.** The features are processed through a linear layer and a depthwise separable convolution, and then passed into the Selective Scan 2D (SS2D) module (as illustrated in Appendix A.2), enhancing spatial modeling through scanning from four diagonal directions.

$$F_{SS2D}^{(2)} = L \left( S \left( D_c \left( L(F') \right) \right) \right) \tag{8}$$

  where $D_c$ denotes depthwise convolution; $S$ denotes Selective Scan 2D module (SS2D).

- **Linear-SiLU Path.** The features are passed through a linear layer followed by a SiLU activation function, introducing non-linearity to enhance model capacity:

$$F_{LS}^{(2)} = \text{SiLU}(L(F')) \tag{9}$$

The outputs of both paths are fused via element-wise multiplication, and then passed through a linear layer to generate the final output of the Mamba branch:

$$F_m{}^{(2)} = L \left( F_{SS2D}^{(2)} \otimes F_{LS}^{(2)} \right), \tag{10}$$

Finally, the outputs of the convolutional and Mamba branches are concatenated along the channel dimension and passed through a channel shuffle operation to promote information interaction and reduce redundancy. The final output $F_m$ of the module is:

$$F_m = C_s \left( [F_c{}^{(1)}, F_m{}^{(2)}] \right), \tag{11}$$

where $C_s$ denotes channel shuffle operation.

**Bidirectional Aware Attention Module(BAAM)** To facilitate bidirectional interactive updates between visual and textual features, we propose the BAAM. Given the visual feature map $F_m \in \mathbb{R}^{C \times H \times W}$ output by the CMamba module and the attribute prototypes $T \in \mathbb{R}^{K \times C}$, we firstly reshape the $F_m$ into a sequence $V \in \mathbb{R}^{N \times C}$ with $N = H \times W$.

A shared pre-projection layer $\phi(\cdot)$ is then applied to both modalities to generate query, key, and value embeddings:

$$\begin{aligned} Q_t, K_t, V_t &= \phi(T) \in \mathbb{R}^{K \times C}, \\ Q_v, K_v, V_v &= \phi(V) \in \mathbb{R}^{N \times C}, \end{aligned} \tag{12}$$

Subsequently, bidirectional cross-modal attention maps between the visual and attribute modalities are computed as:

$$A_t = \text{Softmax}\left(\frac{Q_t K_v^\top}{\sqrt{C}}\right) \in \mathbb{R}^{K \times N},$$
$$A_v = \text{Softmax}\left(\frac{Q_v K_t^\top}{\sqrt{C}}\right) \in \mathbb{R}^{N \times K}. \tag{13}$$

The two modalities are then updated through cross-attention followed by a shared post-projection layer $\psi(\cdot)$:

$$\tilde{T} = \psi(A_t V_v) \in \mathbb{R}^{K \times C},$$
$$\tilde{V} = \psi(A_v V_t) \in \mathbb{R}^{N \times C}. \tag{14}$$

To preserve the original feature information, residual connections are employed:

$$T_u = T + \tilde{T}, \quad F_u = F_m + \tilde{F}. \tag{15}$$

where $\tilde{F}$ denotes the reshaped version of $\tilde{V}$.

To complement the deep semantic features with local details, we fuse the visual features $F_i (i = 1, 2, 3)$ from early layers of ResNet-101 with the deep feature $F_u$ using $1 \times 1$ convolutions, enhancing feature expressiveness and diversity. The fused feature map is then passed through a Softmax function and multiplied with $F_{hg}$ enhanced by the HGA, resulting in the highly discriminative attribute feature $F_a$.

### 3.3.3 Prototype Learning Module

To address semantic ambiguity and domain shift, we define the prototype contrastive loss as a combination of prototype alignment loss and attribute contrastive loss.

**Prototype Alignment Loss** The prototype alignment loss encourages each attribute feature $F_a^i$ to be close to its corresponding prototype $t_i$ and far from other prototypes $t_{i'}$ ($i' \neq i$). It is defined as:

$$\mathcal{L}_{PA} = \sum_{i=1}^{\bar{k}} \text{ReLU}\big(\cos(F_a^i, t_i) - \mu \min_{i' \neq i} \cos(F_a^i, t_{i'})\big), \tag{16}$$

where $\cos()$ is cosine similarity, $\mu$ is a margin (e.g., 0.5), and $\bar{k}$ is the number of valid attributes. The ReLU applies penalty only when the margin condition is not met, enhancing alignment robustness.

**Attribute Contrastive Loss** To improve attribute embedding discrimination, we propose the attribute contrastive loss, which enhances intra-class compactness and inter-class separability by focusing on hard samples. For each attribute feature $F_a^i$, we select a refined set $F_a = \{F_a^i\}_{i=1}^{\bar{k}}$ and construct hard positives $\{F_{i,u}^+\}_{u=1}^U$ (same semantic label, low similarity) and hard negatives $\{F_{i,v}^-\}_{v=1}^V$ (different label, high similarity). Define temperature-scaled similarity $S(a, b) = \exp\big(\cos(a, b)/\tau\big)$. Here $\tau$ is a temperature hyperparameter used to control the sensitivity of the similarity distribution. The contrastive loss for each attribute feature $t_i$ is defined as:

$$\mathcal{L}_{AC} = \frac{1}{\bar{k}} \sum_{i=1}^{\bar{k}} \left(-\frac{1}{U} \sum_{u=1}^U \log \frac{S(F_a^i, F_{i,u}^+)}{\sum_{u=1}^U S(F_a^i, F_{i,u}^+) + \sum_{v=1}^V S(F_a^i, F_{i,v}^-)}\right), \tag{17}$$

The overall Prototype Contrastive Loss is:

$$\mathcal{L}_{PC} = \lambda_1 \mathcal{L}_{PA} + \lambda_2 \mathcal{L}_{AC}, \tag{18}$$

where $\lambda_1, \lambda_2$ balance the two components.

### 3.4 Optimization and Zero-Shot Recognition

#### 3.4.1 Optimization.

We apply global average pooling to $F_{hg}$ to obtain the compact representation $F_g \in \mathbb{R}^C$. To align $F_g$ with the class prototypes $P^c$, we minimize the following cross-entropy loss:

$$\mathcal{L}_{CLS} = -\frac{1}{B} \sum_{i=1}^{B} \log \frac{\exp\left(\alpha \cdot \cos(F_g(x_i), P^c)\right)}{\sum_{\bar{c} \in \mathcal{C}^s} \exp\left(\alpha \cdot \cos(F_g(x_i), P^{\bar{c}})\right)}, \tag{19}$$

where $\alpha > 0$ is a scaling factor, $B$ is the batch size, and $\cos(\cdot, \cdot)$ denotes cosine similarity.

The overall objective of the proposed UMZero model is:

$$\mathcal{L} = \mathcal{L}_{CLS} + \mathcal{L}_{PC}, \tag{20}$$

### 3.4.2 ZERO-SHOT RECOGNITION.

During testing, given an input image $x$, we extract its global feature $F_g(x)$ and evaluate similarity scores with all class prototypes. The predicted label is obtained as

$$\hat{c} = \underset{c \in C^u / \in C^u \cup C^s}{\arg\max} \alpha \cdot \cos(F_g(x), P^c) - \gamma \cdot \mathbb{I}[c \in C^s], \tag{21}$$

where $C^u$ and $C^s$ denote the unseen and seen class sets, respectively. The indicator function $\mathbb{I}[\cdot]$ equals 1 if $c \in C^s$, and 0 otherwise. The notation indicates CZSL when $c \in C^u$, and GZSL when $c \in C^u \cup C^s$. To reduce bias toward seen classes in GZSL, we apply Calibrated Stacking (CS) (Chao et al., 2016) with a calibration factor $\gamma$.

## 4 EXPERIMENTS

### 4.1 EXPERIMENTAL SETUP

#### 4.1.1 DATASETS.

To validate the performance of our proposed UMZero approach, we evaluate it on three standard benchmark datasets: two fine-grained datasets (CUB (Welinder et al., 2010) and SUN (Patterson & Hays, 2012)) and one coarse-grained dataset (AWA2 (Xian et al., 2018)). For a fair comparison, we adhere to the Protocol of Proposed Split (PS) as defined in Xian et al. (2018) to partition seen and unseen classes.

#### 4.1.2 EVALUATION PROTOCOLS.

We follow the evaluation methodology from Xian et al. (2018), employing Top-1 accuracy as the primary metric for ZSL. In the CZSL setting, we report accuracy $ACC$ on the unseen classes alone. In the GZSL scenario, the model is required to recognize both seen and unseen classes. We summarize performance using the harmonic mean $H = \frac{2 \times S \times U}{S+U}$, $S$ and $U$ denote the Top-1 accuracies on seen and unseen classes, respectively.

#### 4.1.3 IMPLEMENTATION DETAILS.

The imlementation deatils are provided in Appendix A.2.

### 4.2 COMPARISON WITH STATE-OF-THE-ART METHODS

We evaluate UMZero under the CZSL protocol on three standard benchmarks. Table 1 compares our method against leading CNN-based (e.g., TransZero++ (Chen et al., 2022a), CREST (Huang et al., 2024)), Transformer-based (e.g., ZSLViT (Chen et al., 2024b), DUET (Chen et al., 2023c)), and Mamba-based (e.g., ZeroMamba (Hou et al., 2025)) methods. UMZero achieves top-1 accuracies of 80.5% (CUB), 68.5% (SUN), and 76.7% (AWA2), surpassing prior methods. Against CNN-based baselines, UMZero improves by 1.9%, 0.9%, and 3.2% on CUB, SUN, and AwA2, respectively; compared to Transformer-based methods, the improvements are 1.6%, 0.2%, and 6.0%. Compared to ZeroMamba, UMZero outperforms by 0.5% on CUB and 4.8% on AWA2, with a minor 3.9% drop on SUN. These gains stem from UMZero's effective fusion of CNN's local feature extraction and CMamba's long-range dependency modeling, further enhanced by the BAAM improving visual-semantic alignment. In the GZSL setting (also in Table 1), UMZero attains the highest harmonic

Table 1: Results (%) from the most advanced CZSL and GZSL techniques on CUB, SUN, and AWA2 datasets are showcased. The top and second-best achievements are highlighted in Red and Blue, respectively. The absence of results is indicated by the symbol "–". * indicates methods based on ViT-Base, and †denotes methods based on VMamba-Small.

| Methods | CUB CZSL ACC | CUB GZSL U | CUB GZSL S | CUB GZSL H | SUN CZSL ACC | SUN GZSL U | SUN GZSL S | SUN GZSL H | AWA2 CZSL ACC | AWA2 GZSL U | AWA2 GZSL S | AWA2 GZSL H |
|---|---|---|---|---|---|---|---|---|---|---|---|---|
| DAZLE (Huynh & Elhamifar, 2020) | 66.0 | 56.7 | 59.6 | 58.1 | 59.4 | 52.3 | 24.3 | 33.2 | 67.9 | 60.3 | 75.7 | 67.1 |
| DPPN (Wang et al., 2021) | - | 70.2 | 77.1 | 73.5 | - | 47.9 | 35.8 | 41.0 | - | 63.1 | 86.8 | 73.1 |
| CLIP* (Radford et al., 2021) | - | 55.2 | 54.8 | 55.0 | - | - | - | - | - | - | - | - |
| GEM-ZSL (Liu et al., 2021) | 77.8 | 64.8 | 77.1 | 70.4 | 62.8 | 38.1 | 35.7 | 36.9 | 67.3 | 64.8 | 77.5 | 70.6 |
| APN (Xu et al., 2022) | 72.0 | 65.3 | 69.3 | 67.2 | 61.6 | 41.9 | 34.0 | 37.6 | 68.4 | 57.1 | 72.4 | 63.9 |
| MSDN (Chen et al., 2022c) | 76.1 | 68.7 | 67.5 | 68.1 | 65.8 | 52.2 | 34.2 | 41.3 | 70.1 | 62.0 | 74.5 | 67.7 |
| TransZero (Chen et al., 2022b) | 76.8 | 69.3 | 68.3 | 68.8 | 65.6 | 52.6 | 33.4 | 40.8 | 70.1 | 61.3 | 82.3 | 70.2 |
| HAS (Chen et al., 2023b) | 76.5 | 69.6 | 74.1 | 71.8 | 63.2 | 42.8 | 38.9 | 40.8 | 71.4 | 63.1 | 87.3 | 73.3 |
| ICIS (Christensen et al., 2023) | 60.6 | 45.8 | 73.7 | 56.5 | 51.8 | 45.2 | 25.5 | 32.7 | 64.6 | 35.6 | 93.3 | 51.6 |
| TransZero++ (Chen et al., 2023a) | 78.3 | 67.5 | 73.6 | 70.4 | 67.6 | 48.6 | 37.8 | 42.5 | 72.6 | 64.6 | 82.7 | 72.5 |
| CoAR-ZSL (Du et al., 2023) | 79.2 | 70.9 | 77.3 | 74.0 | 66.7 | 50.6 | 38.0 | 43.4 | 74.1 | 68.1 | 79.1 | 73.2 |
| DUET* (Chen et al., 2023c) | 72.3 | 62.9 | 72.8 | 67.5 | 64.4 | 45.7 | 45.8 | 45.8 | 69.9 | 63.7 | 84.7 | 72.7 |
| I2MVFormer* (Naeem et al., 2023) | 42.1 | 32.4 | 63.1 | 42.8 | – | – | – | – | 73.6 | 66.6 | 82.9 | 73.8 |
| GNDAN (Chen et al., 2024a) | 75.1 | 69.2 | 69.6 | 69.4 | 65.3 | 50.0 | 34.7 | 41.0 | 71.0 | 60.2 | 80.8 | 69.0 |
| CREST (Huang et al., 2024) | 78.6 | 71.1 | 72.4 | 71.7 | 66.3 | 50.4 | 39.8 | 43.2 | 73.5 | 63.9 | 87.5 | 74.1 |
| ZSLViT* (Chen et al., 2024b) | 78.9 | 69.4 | 78.2 | 73.6 | 68.3 | 45.9 | 48.4 | 47.3 | 70.7 | 66.1 | 84.6 | 74.2 |
| PFRN (Hu et al., 2025) | 77.1 | 72.7 | 75.0 | 73.8 | 66.3 | 55.5 | 32.3 | 40.9 | 71.3 | 68.6 | 84.3 | 75.6 |
| AENET-Res (Ge et al., 2025) | 78.1 | 71.5 | 73.0 | 72.2 | 66.7 | 49.9 | 36.4 | 42.1 | 73.8 | 67.5 | 79.8 | 73.2 |
| ZeroMamba† (Hou et al., 2025) | 80.0 | 72.1 | 76.4 | 74.2 | 72.4 | 56.5 | 41.4 | 47.7 | 71.9 | 67.9 | 87.6 | 76.5 |
| **UMZero(Ours)** | 80.5 | 73.3 | 77.9 | 75.5 | 68.5 | 52.1 | 44.1 | 47.8 | 76.7 | 69.8 | 89.6 | 78.4 |

mean (H): 75.5% (CUB), 47.8% (SUN), and 78.4% (AwA2). Compared to heavier ViT-based models (e.g., ZSLViT (Chen et al., 2024b), I2DFormer (Naeem et al., 2023)), UMZero achieves superior results with a lighter backbone. Compared with ZeroMamba (Hou et al., 2025), UMZero's harmonic mean is higher by 1.3%, 0.1%, and 1.9% on CUB, SUN, and AwA2, respectively. Notably, UMZero surpasses large vision-language models like CLIP (Radford et al., 2021) by 20.5% on CUB's harmonic mean, demonstrating enhanced cross-domain transferability via richer visual representations and stronger visual-semantic consistency.

### 4.3 ABLATION STUDY AND ANALYSIS

#### 4.3.1 COMPONENT ANALYSIS.

To assess the individual impact of each module within UMZero, we perform ablation studies on CUB and SUN (Table 2). Starting from a CNN-only baseline with standard classification loss, we progressively add our proposed modules. Incorporating the BAAM and $L_{PC}$ yields improvements in the harmonic mean of 0.5% and 0.8% on CUB, and 0.5% and 0.3% on SUN, confirming their roles in enhancing visual-semantic alignment and compactness. Adding the HGA further improved global semantic consistency, boosting the H scores to 75.0% on CUB and 46.2% on SUN. Finally, the CMamba integration achieves the best results, with CZSL accuracies of 80.5% and 68.5%, and harmonic means of 75.5% and 47.8% on CUB and SUN, respectively. The consistent degradation observed when removing any single component further validates the complementary contributions of all modules to discriminative power and generalization.

Table 2: Performance Comparison of Different Module Combinations on CUB and SUN Datasets.

| Baseline | BAAM | $L_{PC}$ | HGA | CMamba | CUB CZSL | CUB U | CUB S | CUB H | SUN CZSL | SUN U | SUN S | SUN H |
|---|---|---|---|---|---|---|---|---|---|---|---|---|
| ✓ | ✗ | ✗ | ✗ | ✗ | 77.4 | 66.6 | 79.2 | 72.4 | 65.9 | 46.9 | 41.0 | 43.8 |
| ✓ | ✓ | ✗ | ✗ | ✗ | 77.3 | 67.2 | 79.6 | 72.9 | 66.7 | 49.3 | 40.2 | 44.3 |
| ✓ | ✓ | ✓ | ✗ | ✗ | 77.9 | 71.1 | 76.5 | 73.7 | 66.3 | 50.3 | 40.0 | 44.6 |
| ✓ | ✓ | ✓ | ✓ | ✗ | 78.6 | 71.4 | 78.9 | 75.0 | 66.9 | 50.0 | 42.9 | 46.2 |
| ✓ | ✓ | ✗ | ✓ | ✓ | 79.6 | 70.5 | 79.2 | 74.6 | 67.9 | 48.9 | 43.1 | 45.8 |
| ✓ | ✓ | ✓ | ✗ | ✓ | 79.1 | 71.5 | 76.9 | 74.1 | 67.1 | 52.6 | 39.7 | 45.3 |
| UMZero (full) | ✓ | ✓ | ✓ | ✓ | 80.5 | 73.3 | 77.9 | 75.5 | 68.5 | 52.1 | 44.1 | 47.8 |

### 4.3.2 ANALYSIS OF HYPERPARAMETER.

To evaluate the impact of the hyperparameter $\alpha$, we conducted a sensitivity analysis over the range $10, 15, 20, 25, 30$. As shown in Fig. 3(a-b), under both CZSL and GZSL settings, ACC and H steadily improve as $\alpha$ increases from 10 to 25, then slightly decline at 30. This trend indicates that $\alpha$ plays a key role in balancing attribute discriminability and generalization. Therefore, we fix $\alpha = 25$ in all subsequent experiments. Furthermore, we investigate the influence of the other key hyperparameters of UMZero on the CUB dataset: the prototype alignment loss weight $\lambda_1$, the attribute contrastive loss weight $\lambda_2$, and the temperature coefficient $\tau$. As illustrated in Fig. 3(c-e), UMZero attains its best performance in terms of both ACC and H when $\lambda_1 = 0.1$, $\lambda_2 = 0.8$, and $\tau = 0.1$. More results are placed in the Appendix A.3.

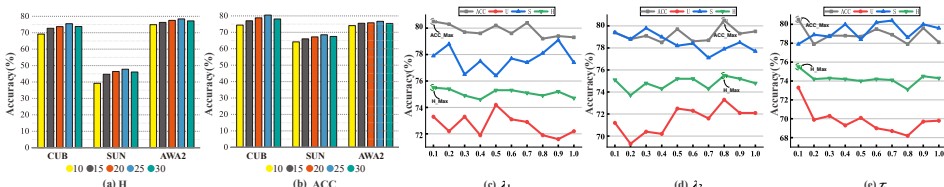

Figure 3: Performance variation and sensitivity analysis. Performance variation with respect to different $\alpha$ values on three datasets under CZSL (ACC) and GZSL (H) settings (a-b), as well as the performance sensitivity to hyperparameters on the CUB dataset (c-e).

### 4.4 QUALITATIVE RESULTS

### 4.4.1 VISUALIZATION OF T-SNE EMBEDDINGS.

To evaluate UMZero's ability to distinguish attribute features, we visualized 15 randomly selected attributes from the CUB and SUN datasets using t-SNE (see Fig. 4). Compared to the baseline's scattered and overlapping features, UMZero produces more compact and clearly separated clusters, demonstrating stronger intra-class consistency and inter-class separability.

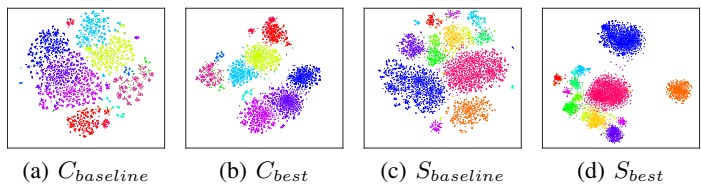

(a) $C_{baseline}$     (b) $C_{best}$     (c) $S_{baseline}$     (d) $S_{best}$

Figure 4: The visualization of attribute features using T-SNE on CUB and SUN datasets.(In the figure, C and S represent the CUB and SUN datasets, respectively. )

## 5 CONCLUSION

This paper proposes a novel ZSL framework, UMZero, which achieves deep integration of fine-grained visual information and global semantic representations. Specifically, the local features extracted by the backbone network are optimized by the high-order global aggregator (HGA) and aggregated into a global dynamic space modeled by the mamba interaction module (MIM), thereby enhancing detail perception and semantic completeness. Meanwhile, MIM establishes a bidirectional interaction pathway between the visual and semantic modalities, facilitating cross-modal matching and improving feature alignment consistency. On this basis, a prototype contrastive loss is introduced to optimize inter-class separability and intra-class compactness at the prototype level, effectively alleviating issues such as information redundancy and semantic shift. Experimental results demonstrate that UMZero significantly outperforms existing methods on three widely used benchmark datasets.

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

## A APPENDIX

### A.1 PRELIMINARIES OF SSMs

Structured State Space Models (SSMs) are effective for sequence modeling, offering robust global context with linear computational complexity. These models draw inspiration from continuous-time dynamical systems, wherein the input signal is embedded into a latent state space to enable efficient, expressive temporal reasoning. In continuous time, a general SSM is governed by a linear ordinary differential equation (ODE):

$$
\begin{aligned}
h'(t) &= \mathbf{A}h(t) + \mathbf{B}x(t), \\
y(t) &= \mathbf{C}h(t),
\end{aligned}
\tag{22}
$$

where $x(t) \in \mathbb{R}$ denotes the input, $h(t) \in \mathbb{R}^N$ is the $N$-dimensional latent state, and $y(t) \in \mathbb{R}$ is the output. The matrices $\mathbf{A} \in \mathbb{R}^{N \times N}$, $\mathbf{B} \in \mathbb{R}^{N \times 1}$, and $\mathbf{C} \in \mathbb{R}^{1 \times N}$ represent the state-transition, input-projection, and output-projection operators, respectively.

To apply this model in discrete-time deep learning systems, the continuous dynamics are discretized using the Zero-Order Hold (ZOH) assumption. For a time step $\Delta \in \mathbb{R}$, the discrete equivalents of system matrices are computed as:

$$
\begin{aligned}
\overline{\mathbf{A}} &= \exp(\Delta \mathbf{A}), \\
\overline{\mathbf{B}} &= (\Delta \mathbf{A})^{-1} \left( \exp(\Delta \mathbf{A}) - \mathbf{I} \right) \Delta \mathbf{B},
\end{aligned}
\tag{23}
$$

where $\exp(\cdot)$ denotes the matrix exponential and $\mathbf{I}$ is the identity matrix. This transformation allows efficient implementation in modern neural architectures.

The resulting discrete-time state dynamics are defined by:

$$h_t = \overline{\mathbf{A}}h_{t-1} + \overline{\mathbf{B}}x_t,$$
$$y_t = \overline{\mathbf{C}}h_t, \tag{24}$$

For long sequence modeling, SSMs enable efficient global dependency capture through a structured convolution kernel. By recursively unrolling the state transitions, a length-$L$ convolution kernel $\overline{\mathbf{K}} \in \mathbb{R}^L$ can be constructed as:

$$\overline{\mathbf{K}} = \left( \mathbf{C}\overline{\mathbf{B}}, \ \mathbf{C}\overline{\mathbf{A}}\overline{\mathbf{B}}, \ \ldots, \ \mathbf{C}\overline{\mathbf{A}}^{L-1}\overline{\mathbf{B}} \right), \tag{25}$$

which defines the linear time-invariant convolutional representation:

$$y = x * \overline{\mathbf{K}}, \tag{26}$$

## A.2 SS2D

Selective Scan 2D (SS2D) module consists of scan extension, an S6 module, and scan fusion (as illustrated in Fig. 5), enhancing spatial modeling through scanning from four diagonal directions.

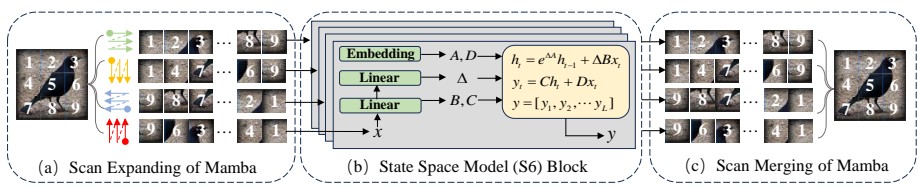

(a) Scan Expanding of Mamba  (b) State Space Model (S6) Block  (c) Scan Merging of Mamba

Figure 5: Overview of the SS2D Architecture.

## A.3 PARAMETER ANALYSIS RESULTS ON THE SUN AND AWA2

To further verify the robustness of UMZero across different datasets, we conduct a hyperparameter ablation study on the AWA2 and SUN datasets, examining the impact of the prototype alignment loss weight $\lambda_1$, the attribute contrastive loss weight $\lambda_2$, and the temperature coefficient $\tau$ on model performance. As illustrated in Fig. 6, UMZero achieves the best performance on AWA2 and SUN when $\lambda_1$ is set to 1.0 or 0.7, $\lambda_2$ is set to 0.6 or 0.5, and $\tau$ is 0.1.

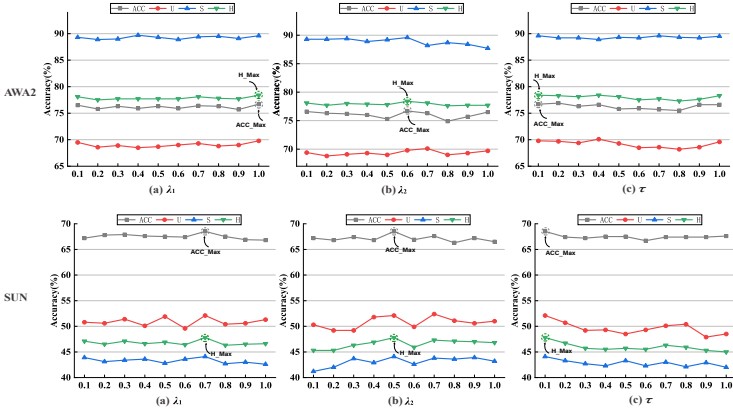

Figure 6: Performance Sensitivity to Hyperparameters on the AWA2 and SUN

