# OpenReview forum: "UMZero: A Unified CNN-Mamba Framework for Zero-Shot learning"
_ICLR.cc/2026/Conference — ICLR 2026 Conference Withdrawn Submission_

### Official Review · Reviewer_zXHZ · 2025-10-16

**Soundness:** 2
**Presentation:** 2
**Contribution:** 2
**Rating:** 4
**Confidence:** 4

**Summary:**

This paper proposes a unified CNN-Mamba framework, termed UMZero, for advancing Zero-Shot Learning (ZSL). To fully exploit complex cross-modal interactions, the paper presents a hybrid ZSL method to combine the capability of CNN and state space model (SSM), which integrates the High-order Global Aggregator (HGA), the Mamba Interaction Module (MIM), and a prototype learning unit. Experiment on three ZSL datasets demonstrates its performance and generalization capabilities.

**Strengths:**

1. The paper proposes the first hybrid ZSL framework, which combines a pretrained CNN with an SSM.
2. The idea of HGA is novel and noteworthy.
3. The proposed approach outperforms other methods on different datasets.

**Weaknesses:**

1. The paper is not well-established. I've identified a number of incorrect figure references that affect clarity.
2. The biggest concern regarding the paper is the performance of UMZero. While the paper claims that the hybrid framework can capture more discriminative features for fine-grained attribute modeling, it only marginally outperforms ZeroMamba on CUB by 0.5% and severely underperforms on SUN. It lacks in-depth analysis in this paper.
3. The scalability of UMZero is unverified, it lacks the large-scale ImageNet benchmark reported by ZeroMamba.
4. To demonstrate the effectiveness of the proposed method, the paper should make comparison with latest approaches, such as VADS [1], ZeroDiff [2], SVIP [3].


Reference:

[1] (2024 CVPR) Visual-Augmented Dynamic Semantic Prototype for Generative Zero-Shot Learning

[2] (2025 ICLR) ZeroDiff: Solidified Visual-Semantic Correlation in Zero-Shot Learning

[3] (2025 ICCV) SVIP: Semantically Contextualized Visual Patches for Zero-Shot Learning

**Questions:**

My concerns are listed in the ''Weaknesses".

---

### Official Review · Reviewer_H5bk · 2025-10-27

**Soundness:** 3
**Presentation:** 2
**Contribution:** 2
**Rating:** 4
**Confidence:** 4

**Summary:**

Addressing the core challenge in Zero-Shot Learning (ZSL)—the difficulty of jointly modeling local features and global correlations—this paper proposes UMZero, a hybrid framework that represents the first ZSL solution fusing a pre-trained CNN with Mamba (a State Space Model, SSM). The framework resolves limitations of existing methods through three core modules: 1) the High-order Global Aggregator (HGA), which enhances feature expressiveness by capturing high-order statistical dependencies across channels; 2) the Mamba Interaction Module (MIM), which combines the fine-grained local extraction capability of CNNs with Mamba’s long-range dependency modeling to enable cross-modal bidirectional information flow and alignment; and 3) the prototype learning module, which introduces a prototype contrastive loss to construct a semantically structured embedding space, improving intra-class compactness and inter-class discriminability.

The authors validated UMZero’s performance on three mainstream ZSL datasets (CUB, SUN, and AWA2).

**Strengths:**

1. UMZero is the first to deeply fuse the local feature extraction advantages of CNNs with the efficient global modeling capabilities of Mamba.
2. The bidirectional cross-modal interaction mechanism (BAAM component) in the MIM module fills the gap in existing ZSL models, which "only rely on semantics to guide visual updates and lack bidirectional visual-semantic feedback." This allows semantic prototypes to be dynamically optimized based on visual information, effectively bridging the representational gap between visual and semantic spaces.

**Weaknesses:**

1. Insufficient depth of innovation and ambiguous boundary of innovation in CNN-Mamba fusion. Although the authors claim UMZero is the "first CNN-Mamba fused ZSL framework," they do not fully compare it with related work on "CNN-SSM fusion in the visual domain" (e.g., variants of Vision Mamba (Zhu et al., 2024)). The essential differences between UMZero’s fusion strategy (e.g., feature splitting, channel shuffle, bidirectional interaction) and these works are not clearly explained, which may weaken the uniqueness of its innovation.
2. Omission of key baselines in related work. In the current ZSL field, methods based on "bidirectional cross-modal interaction" (e.g., CoAR-ZSL (Du et al., 2023) in 2023 and GNDAN (Chen et al., 2024a) in 2024) have also attempted to address visual-semantic alignment. However, the authors do not deeply analyze the differences in interaction mechanisms between UMZero’s BAAM component and these methods in both related work and experimental comparisons, resulting in insufficient horizontal comparison support for the innovation of "bidirectional interaction."
3. Vague explanation of core mechanisms in the MIM module. The authors mention that the CMamba branch of MIM splits features into two parts (for input to convolution and Mamba respectively) through "channel splitting," but the basis for selecting the "splitting ratio" (e.g., 1:1 channel split) is not explained, nor is the impact of the splitting ratio on performance analyzed. Additionally, the BAAM component only provides the final update equations (Equations (7)–(10)) for "bidirectional cross-modal attention calculation," without detailed derivation of the attention weight calculation process, reducing the reproducibility of the key module.
4. Inadequate theoretical justification for the prototype contrastive loss. The authors define the prototype contrastive loss as a combination of "prototype alignment loss + attribute contrastive loss," but do not theoretically explain why this combination can alleviate "domain shift." For example, there is no comparison with other loss functions in the ZSL field (e.g., contrastive loss (Chen et al., 2022b), triplet loss (Wang et al., 2021)), nor is the basis for selecting the optimal ratio of λ₁ and λ₂ analyzed, leaving the design of the loss function without theoretical support.
5. Insufficient depth of experimental design and robustness validation. When ablating "CMamba," only the performance with and without CMamba is compared, and the individual contributions of the "convolution branch" and "Mamba branch" are not decomposed. This makes it impossible to clarify their specific roles in local-global modeling, weakening the persuasiveness of the ablation conclusions. The selection of hyperparameters is based on empirical choices from the results, and the authors did not explain the reasons for choosing each parameter in this manner.
6. Many images and chapters in the text do not correspond accurately, and the writing expression and formula explanations are also insufficiently precise. For example: “In UMZero (Fig. 6), we integrate a hybrid CNN–Mamba architecture for ZSL.” “The imlementation deatils are provided in Appendix A.2.”

**Questions:**

1. The authors state that UMZero is the "first CNN-Mamba fused ZSL framework," but Vision Mamba (Zhu et al., 2024) has already attempted to apply Mamba to visual tasks, and some works (e.g., ZeroMamba) have introduced Mamba into ZSL. We ask: 1) What are the essential differences between UMZero’s CNN-Mamba fusion strategy (e.g., feature splitting, channel shuffle) and Vision Mamba’s "Mamba adapted for vision"? 2) Why was "CNN + Mamba" chosen instead of "ViT + Mamba" fusion? Is there experimental or theoretical evidence to prove that the advantage of CNNs in local feature extraction provides a more significant gain for ZSL than ViTs?
2. The "Baseline" in the ablation study does not clearly state whether it includes a basic semantic alignment module. If the Baseline is only a pure CNN classification model, it may underestimate the baseline performance of existing methods.
3. How was the "channel splitting ratio" of the MIM module determined? Are there experiments to verify the impact of different splitting ratios (e.g., 1:1, 1:2, 2:1) on performance?

---

### Official Review · Reviewer_BL28 · 2025-10-31

**Soundness:** 3
**Presentation:** 3
**Contribution:** 2
**Rating:** 6
**Confidence:** 3

**Summary:**

This paper proposed a novel framework for zero-shot learning (both the conventional task and the generalized version). The authors pointed out that current CNN-based methods often overlook long-range dependencies due to their finite inceptive field, while state space models (SSM) can't capture fine-grained features. To overcome these challenges, the authors proposed a hybrid model which combines the strengths of CNN and SSM to fully utilize both local fine-grained features and global context. The motivation is straightforward, the model architecture is clear, and the performance looks good.

**Strengths:**

1. The overall design fully utilized both local and global information with different architectures. Specifically, the high-order global aggregator (HGA) enhanced feature compactness and cross-channel dependencies by capturing the second-order channel statistics; the convolutional-mamba module captured both fine-grained local features and spatial dependencies by combining the advantages of CNN-based structure and SSMs; the bidirectional aware attention modules helped to capture the interactions between visual and text features.
2. Besides a commonly used prototype alignment loss, the proposed attribute contrastive loss encouraged intra-class compactness and inter-class separability, providing a clearer prototype space for classification.
3. The performance on both CZSL and GZSL outperformed current SOTA methods under most circumstances, showcasing the effectiveness of the proposed method, and the functionalities of different components were supported by their ablation studies.

**Weaknesses:**

Some concerns on the visualization analysis:
1. It is unclear which baseline is used. It would be better if more than one baseline is compared here.
2. The authors only showed the distribution of the selected features. Based on existing works, the authors might also consider showing the separability of the learned semantic representations from different classes.

**Questions:**

1. I would suggest the authors to pay attention to their figure citations. For example, in line 74 and line 179, I believe the citation should be Fig. 2 instead of Fig. 6. It is recommended to check more details in their writing.
2. In Section 2, line 146-148, the sentences "Nevertheless, its implicit treatment of local regions constrains its fine-grained attribute modeling. However, its performance in fine-grained attribute modeling remains limited." are redundant.

---

### Official Review · Reviewer_7XYH · 2025-11-01

**Soundness:** 3
**Presentation:** 3
**Contribution:** 2
**Rating:** 2
**Confidence:** 5

**Summary:**

UMZero is a hybrid ZSL framework combining CNN and state space models to capture both local and global features. Its core includes a High-order Global Aggregator for channel dependencies, a Mamba Interaction Module for cross-modal fusion, and prototype learning for discriminative representations, effectively solving cross-modal interaction challenges.

**Strengths:**

UMZero's architecture comprises three specialized modules: the High-order Global Aggregator (HGA) enhances features through cross-channel dependency modeling; the Mamba Interaction Module (MIM) enables bidirectional local-global cross-modal fusion; and the prototype learning unit constructs a discriminative semantic embedding space. Together, these components effectively address cross-modal interaction challenges in zero-shot learning. The integrated modules significantly strengthen cross-modal representation learning. Besides, decent performance is obtained on three benchmarks.

**Weaknesses:**

Despite the interesting contributions, I have several significant concerns regarding the manuscript:
1. The manuscript justifies the use of Mamba over Transformer by citing its linear complexity. However, this claim lacks empirical support. To substantiate this design choice, it is essential to include an efficiency analysis comparing GPU memory, inference time, number of parameters, and GFLOPs. Furthermore, a performance comparison with a self-attention-based baseline is necessary to validate the advantage of the proposed approach.
2. The related work and experiments primarily focus on embedding-based methods. To better position the contribution, the review should be expanded to include more recent generative approaches for zero-shot learning, particularly state-of-the-art methods from 2024-2025.
3. To better interpret the model's behavior, the authors should provide attention visualizations on real samples, illustrating what the model focuses on during prediction.
4. The performance improvement on AWA2 is notably more significant than on CUB and SUN. The authors should analyze and discuss the potential reasons for this dataset-dependent performance variation.
5. The authors should discuss the potential of integrating large language models (LLMs) or vision-language models (VLMs) like CLIP. Exploring how their method could leverage these powerful pre-trained representations would strengthen the discussion and highlight future research directions.

**Questions:**

See above.

---

### Note · Authors · 2026-02-01

I have read and agree with the venue's withdrawal policy on behalf of myself and my co-authors.

---

### Meta-Review · Area_Chair_vyUX · 2026-01-08

**Summary:**

Reviewers agreed that the problem is important and that combining CNNs with Mamba for zero-shot learning is a reasonable idea. However, they felt that the method is not clearly novel enough for a top-tier venue. Key design choices were not well justified, and the experimental validation was not strong enough. In particular, missing comparisons to recent methods, limited efficiency analysis, and unclear differences from existing work outweighed the reported performance gains.

**Reviewer Concerns:**

There is no rebuttal.

**Reviewer Scores:**

There is no rebuttal.

---

### Decision · Program_Chairs · 2026-01-26

Reject